

# Consistent administration of cetuximab is associated with favorable outcomes in recurrent/metastatic head and neck squamous cell carcinoma in an endemic carcinogen exposure area: a retrospective observational study

Hui-Ching Wang[1,2], Pei-Lin Liu[3,4], Pei-Chuan Lo[4], Yi-Tzu Chang[4], Leong-Perng Chan[1,5], Tsung-Jang Yeh[1,2], Hui-Hua Hsiao[2,6] and Shih-Feng Cho[2,6]

[1] Graduate Institute of Clinical Medicine, College of Medicine, Kaohsiung Medical University, Kaohsiung, Taiwan
[2] Division of Hematology and Oncology, Department of Internal Medicine, Kaohsiung Medical University Hospital, Kaohsiung Medical University, Kaohsiung, Taiwan
[3] Faculty of Internal Medicine, Specialist Nursing office, Kaohsiung Medical University Hospital, Kaohsiung Medical University, Kaohsiung, Taiwan
[4] Department of Nursing, Kaohsiung Medical University Hospital, Kaohsiung Medical University, Kaohsiung, Taiwan
[5] Department of Otolaryngology-Head and Neck Surgery, Kaohsiung Medical University Hospital, Kaohsiung Medical University, Kaohsiung, Taiwan
[6] Faculty of Medicine, College of Medicine, Kaohsiung Medical University, Kaohsiung, Taiwan

Corresponding author
Shih-Feng Cho, sfcho@kmu.edu.tw

## ABSTRACT

**Background.** This study aimed to analyze the clinical outcomes associated with patients with recurrent/metastatic head and neck squamous cell carcinoma (RM HNSCC) who received cetuximab-based chemotherapy in a real-world clinical setting.

**Methods.** Clinical data were extracted from RM HNSCC patients diagnosed between 2016 and 2019. Kaplan–Meier survival estimates and Cox proportional hazards model were used for survival analyses.

**Results.** Of 106 RM HNSCC patients (mean age = 55.1 years), 38.7% exhibited recurrent disease and 61.3% had metastatic disease. The majority of patients showed a habit of addictive substance use, including alcohol (67.0%), betel nuts (71.7%), or tobacco (74.5%). The primary tumor sites included the oral cavity (64.1%), hypopharynx (19.8%), and oropharynx (16.0%). The median number of cetuximab cycles for the 106 patients was 11 (2–24). The disease control rate (DCR) was 48.1%, and the overall response rate (ORR) was 28.3%. The median progression-free survival (PFS) and overall survival (OS) were 5.0 and 9.23 months, respectively. Patients treated with more than 11 cycles of cetuximab exhibited a longer median PFS and median OS than did patients treated with less than 11 cycles (median PFS: 7.0 vs. 3.0 months, $p < 0.001$; OS: 12.43 vs. 4.46 months, $p = 0.001$). Patients without previous concurrent chemoradiotherapy (CRT) had a better median PFS than did those with previous CRT (6.0 vs. 4.0 months, $p = 0.046$). Multivariable analysis revealed that perineural invasion and fewer cycles of cetuximab (<11 cycles) were independent risk factors associated with disease progression. In addition, the reduction in treatment cycles of cetuximab

and advanced lymph node metastasis were independent prognostic factors predicting poorer overall survival.

**Conclusion**. Our study provides important real-world data regarding cetuximab-containing treatment in RM HNSCC. Consistent administration of cetuximab could be associated with more favorable outcomes in RM HNSCC in endemic carcinogen exposure areas.

# INTRODUCTION

Head and neck squamous cell carcinoma (HNSCC) is the sixth most common malignancy in the world; recurrent and/or metastatic head and neck squamous cell carcinoma (RM-HNSCC) harbors lethal clinical features and dismal medical outcomes (*Parkin et al., 2005*). Over 90% of head and neck cancers are squamous cell carcinomas that develop from the mucosa of the oral cavity, oropharynx, larynx, or hypopharynx (*Warnakulasuriya, 2009*). In Western countries, a subgroup of oropharyngeal SCC is related to human papillomavirus (HPV) infection (*Gatta et al., 2015*; *Gillison et al., 2000*). However, oral cavity SCC is the most predominant site of head and neck cancer in Taiwan due to high prevalence of betel nut consumption (*Belcher et al., 2014*; *Chang et al., 2017*). Virus-induced HNSCC in Western countries is different from its Taiwanese counterpart in that the mechanism of tumorigenesis of HNSCC in Taiwan is mainly related to carcinogens and addictive substances, including alcohol, betel nuts, and tobacco (*Cancer IAfRo, 2012*). These carcinogen-related HNSCCs harbor higher *Ras* oncogene mutations and increased chromosome instability, suggesting that the genetic background and clinical features may be unique to these patients (*Chang et al., 1991*; *Kuo et al., 1994*; *Riaz et al., 2014*).

Epidermal growth factor receptor (EGFR) is usually upregulated with increased levels of its ligand transforming growth factor alpha (TGF-$\alpha$) in most HNSCCs, with both proteins contributing to the carcinogenesis of HNSCC (*Grandis, 2007*). Upregulation of EGFR is an independent poor prognostic factor in HNSCCs (*Ang, Andratschke & Milas, 2004*; *Dassonville et al., 1993*). Cetuximab, an IgG1 chimeric monoclonal antibody targeting EGFR, was one of the first-line treatments for RM HNSCC patients with low programmed death ligand 1 (PD-L1) expression (*Burtness et al., 2019*; *Vermorken et al., 2008*). The addition of cetuximab to platinum-based chemotherapy with fluorouracil (platinum-fluorouracil) improved the overall response rates, median progression-free survival (PFS), and overall survival (OS) compared with chemotherapy alone. Another combination of cetuximab with chemotherapy agents such as taxane also demonstrated substantial benefits (*Adkins et al., 2018*; *Friesland et al., 2018*; *Guigay et al., 2019*). However, most of these clinical trials were conducted in Western countries with fewer patients with primary oral cavity cancer; data regarding the effect of carcinogens such as betel nuts on outcome are very limited. In addition, the percentage of HPV infection status is quite different between

Asian and Western countries, suggesting distinct tumor microenvironments (*Wang, Chan & Cho, 2019*).

In Taiwan, cetuximab combined with systemic chemotherapy has been indicated as first line treatment in patients with RM HNSCC by the National Health Insurance since 2016. After receiving approval for application, the patients can receive cetuximab-containing treatment without copayment. Because of limited financial resources, cetuximab can only be administered in a total of eighteen cycles if no progression is noted. Unlike clinical trials that provide subjects with maintenance cetuximab, patients in real life cannot afford continuous maintenance with high-cost cetuximab to control their disease. Therefore, modifying the treatment protocol wound be a possible strategy (*Hsu & Lu, 2016*; *Shih et al., 2015*). Nevertheless, the impact of modifications such as limiting cetuximab treatment cycle on patient outcome remains unknown. Moreover, real-world data on cetuximab in RM HNSCC patients with high percentages of exposure to various carcinogen remains are also very limited. To answer these questions, we conducted this retrospective and single-arm study to analyze clinical data, hoping to determine the clinical outcomes and prognostic factors in this subset of RM HNSCC patients.

## MATERIALS AND METHODS

### Patient characteristics

Clinicopathological data of patients with HNSCC were confirmed by pathological examination of specimens from biopsy or surgery, and the positive samples were collected and analyzed. A total of 106 cases of RM HNSCC were identified with metastasis or recurrence and were deemed unsuitable for locoregional curative treatment at the Kaohsiung Medical University Hospital. The inclusion criteria included: age at diagnosis 20 years or older; tumor histology of squamous cell carcinoma (grade 1 to grade 3); ICD-9 site code-specific for the oral cavity (OC), hypopharynx (HPC), oropharynx (OPC), and larynx; and treatment with cetuximab from January 2016 to April 2019. The exclusion criteria included secondary malignancy; tumor histology of carcinoma *in situ*; and SCC of the nasopharynx or salivary glands.

### Study design

This was an observational, retrospective, single-center, single-arm study, and the treatment schema is shown in Fig. 1. The collected medical and demographic data included age, gender, alcohol, betel nut usage, tobacco habits, and other clinical parameters obtained from the medical records or interviews with patients. The clinicopathological factors included types and grade of histology, size of tumor, lymph node status, surgical margin, perineural invasion, lymphovascular invasion, and extranodal extension. We defined CRT (chemoradiotherapy)-refractory patients as patients with disease progression during CRT or within three months of the end of CRT. The primary endpoints were median OS and PFS. Specifically, the median OS and PFS (defined as the time from registration to objective disease progression or death from any cause) were determined after the addition of cetuximab to chemotherapy. Other endpoints included the assessment of treatment response and disease control. This study was approved by the Institutional Review Board
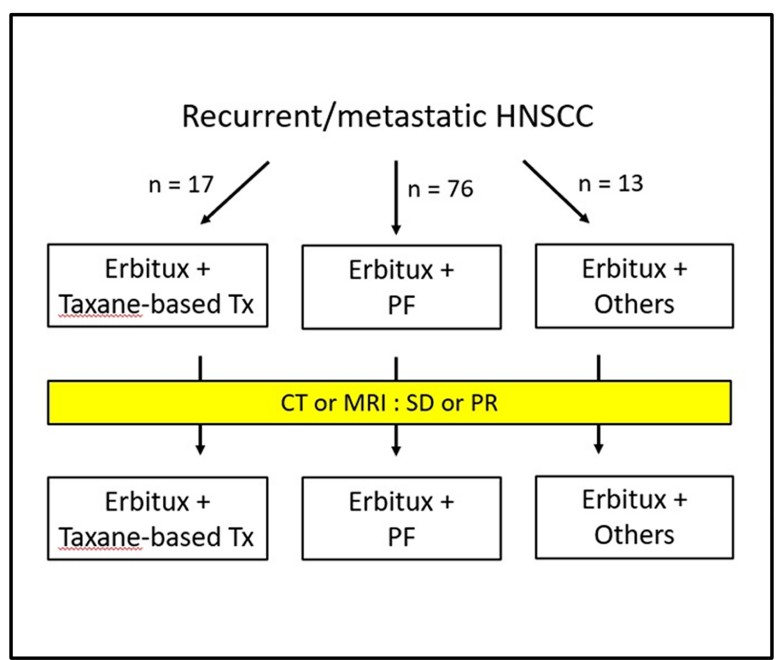

**Figure 1 Treatment Schema.** Tx, treatment; PF, cisplatin and fluorouracil; CT, computed tomography; MRI, magnetic resonance imaging; SD, stable disease; PR, partial response.

and Ethics Committee of Kaohsiung Medical University Hospital (KMUHIRB-E(II)-20190357). The data were analyzed anonymously, and therefore, no additional informed consent was required. All methods were performed in accordance with approved guidelines and regulations.

## Treatment

All patients received cetuximab (250 mg/m$^2$) weekly with a loading dose of 400 mg/m$^2$ until disease progression was noted. The regimen of chemotherapy included PF 75/1000 (cisplatin at 75 mg/m$^2$ or carboplatin at AUC = 5 every 3 weeks plus fluorouracil at 1,000 mg/m$^2$/d for 4 days every 3 weeks), PF 60/800 (cisplatin at 60 mg/m$^2$ or carboplatin at AUC5 every 3 weeks plus fluorouracil at 800 mg/m$^2$/d for 4 days every 3 weeks), taxane-based chemotherapy (docetaxel and cisplatin 75 mg/m$^2$ both at day 1 and every 3 weeks for four courses of paclitaxel 80 mg/m$^2$ weekly), and MTX (methotrexate 40 mg/m$^2$ weekly). The patients could receive chemotherapy or concurrent chemoradiotherapy with weekly cisplatin administration previously before recruitment.

## Treatment response and safety assessment

All patients were followed regularly as outpatients at the medical oncology and department (OPD) of otorhinolaryngology. During the cetuximab treatment period, the patients visited the OPD of medical oncology weekly and that of otorhinolaryngology monthly. The evaluation of disease status included tumor site inspection, laboratory text, and imaging studies. Treatment response was assessed and determined using computed tomography
(CT) or magnetic resonance imaging (MRI) at baseline (before cetuximab) and at 3-month intervals after treatment was started. Imaging within 4 weeks before cetuximab was acceptable, and imaging could be performed whenever clinical physicians suspected disease progression. RECIST version 1.1 was used to determine disease progression and tumor response.

The treatment response of patients was classified into four categories: complete response (CR, disappearance of all target lesions), partial response (PR, decrease in target lesion diameter sum >30%), progression disease (PD, increase in target lesion diameter sum >20%), and stable disease (SD, does not meet other criteria). The calculation of overall response rate (ORR), including patients classified as having complete and partial responses, was based on the best objective response achieved during cetuximab treatment. The calculation of disease control rate (DCR) included patients classified as having complete response, partial response, and stable disease. After disease progression, further treatments and survival status were documented every 3 months. Regarding safety assessment, treatment-related adverse events were monitored weekly throughout the study and were evaluated using Common Terminology Criteria for Adverse Events version 4.0.

## Statistical analysis

The primary goal of the study was to analyze the outcome of cetuximab-based chemotherapy in recurrent or metastatic settings, including a comparison between median PFS and OS among patients receiving various cycles of cetuximab and regimens of chemotherapy. The location of primary sites (OC, OPC, or HPC), histological grade (Grades 1, 2, 3), tumor size and status (T1, T2, T3, T4), lymph node status (N0, N1, N2, N3), stage at initial diagnosis (I, II, III, or IV), surgery status (with or without previous surgery), CRT (with or without previous CRT), and chemotherapy before cetuximab therapy (with or without prior chemotherapy) were all included for analysis. Between-group comparisons were analyzed using Fisher's exact test and Pearson's chi-square test for various categorical variables. We calculated median PFS and OS using Kaplan–Meier analysis, and we analyzed differences between the curves using the log-rank test. We defined the median PFS as the time between the start of disease progression and treatment, including disease progression or death. Patients alive and without disease progression by the final follow-up visit were considered potential right censoring subjects, and the follow-up interval was truncated at the end of study. Univariate and multivariable analyses using the Cox proportional hazard model were preformed to analyze prognostic factors associated with cetuximab treatment. The factors for this analysis included age at initial diagnosis, location of primary sites, histological grade, pathological features (margin, lymphovascular invasion, perineural invasion, and extranodal extension), tumor size, lymph node status, stage at initial diagnosis, previous treatment before cetuximab (surgery, chemotherapy, or CRT), combined regimen and dosage of chemotherapy. All $p$-values were considered significant if $p < 0.05$ and were two-sided. Statistical analyses were performed using STATA version 11 (STATA Corp., TX, USA).

## RESULTS

### Baseline characteristics of patients

Clinical data from 106 patients (99 males and 7 females) with a median age of 55.1 years were collected. Among these patients, 65 patients (61.3%) had metastatic disease and 41 patients (38.4%) had recurrent disease with initiation of cetuximab. Almost all patients had addictions to alcohol or betel nuts or history of smoking, including 61 patients (57.5%) with exposure to all three carcinogens. Only 5 patients (4.7%) had no previous exposure to these risk factors. Regarding the tumor site, most of the primary sites had origins in the oral cavity (64.1%), followed by the hypopharynx (19.8%), and oropharynx (16.0%). The majority of patients had advanced disease, including T3-4, N2-3, or clinical stage 4. The details of basic information of the study population are listed in Table 1.

### Treatment modality

With respect to prior treatment before cetuximab treatment, most patients had undergone various HNSCC treatments, including surgery (78.3%), chemotherapy (81.1%) and CRT (80.2%). In addition, there were 34 CRT-refractory patients who suffered from disease progression during CRT or within three months of the end of CRT.

The major reason for cetuximab treatment was recurrent disease with metastatic tumors. The median number of cycles of cetuximab was 11 (2–24), with 60 patients receiving ≥11 cycles of cetuximab, and 46 patients receiving <11 cycles of cetuximab. Among these patients, 76 patients received chemotherapy with the EXTREME regimen (cisplatin and fluorouracil) and 17 patients received taxane-based chemotherapy. The median number of cetuximab administration cycles in these 76 patients with a PF regimen was 11 (range: 2–24) while the median number of cetuximab cycles in 17 patients using taxane-based regimen was 12 (range: 4–23). There was no significant difference in the number of cetuximab cycles between the two groups ($p = 0.427$). The details of the treatment modalities are shown in Table 2. The demographic data of various cetuximab cycles (≥11 and <11) are shown in Tables S1 and S2. Interestingly, there was no difference in terms of previous treatments, including surgery, chemotherapy, and CRT, between patients who received <11 cycles of cetuximab and those who received ≥11 cycles of cetuximab.

### Treatment outcomes

After cetuximab treatment, clinical responses were observed in 30 patients including 1 complete response and 29 partial responses, with ORR of 28.3%. When the patients with stable disease ($n = 21$, 19.8%) were included in the analysis, the disease control rate was 48.1%. The median PFS and OS were 5 months and 9.23 months, respectively. As of the cut-off date, only one patient did not progress, and 38 patients survived. The median PFS was 5 months (95% CI [3.0–6.0] months) and the median OS was 9.23 months (95% CI [7.03–13.84] months). The treatment responses according to various stages are shown in Table S3.

The median PFS in various subgroups stratified by treatment modalities is shown in Fig. 2. Notably, the patients who received more cetuximab treatment (≥11 cycles) had a better median PFS than did patients who received less cetuximab (7 months vs. 3 months,

**Table 1  Baseline characteristics in the entire cohort ($N = 106$).**

| Variables | n (%) |
|---|---|
| Age, years (mean ± SD) | 55.1 ± 9.9 |
| Alcohol | 71 (67.0%) |
| Betel nuts | 76 (71.7%) |
| Smoking | 79 (74.5%) |
| Primary sites | |
| HPC | 21 (19.8%) |
| OC | 68 (64.1%) |
| OPC | 17 (16.0%) |
| Grade | |
| 1 | 28 (26.4%) |
| 2 | 57 (53.8%) |
| 3 | 16 (15.1%) |
| Unknown | 5 (4.7%) |
| Margin positivity | 11 (10.4%) |
| LVI, positive | 4 (3.8%) |
| PNI, positive | 9 (8.5%) |
| ENE, positive | 5 (4.7%) |
| Tumor size | |
| T0 | 2 (1.9%) |
| T1 | 14 (13.2%) |
| T2 | 24 (22.6%) |
| T3 | 16 (15.1%) |
| T4 | 50 (47.2%) |
| Lymph node status | |
| N0 | 27 (25.5%) |
| N1 | 12 (11.3%) |
| N2 | 56 (52.8%) |
| N3 | 11 (10.4%) |
| Stage at initial diagnosis | |
| I | 9 (8.5%) |
| II | 6 (5.7%) |
| III | 11 (10.4%) |
| IV | 80 (75.5%) |

Notes.

HPC, hypopharyngeal cancer; OC, oral cavity cancer; OPC, oropharyngeal cancer; LVI, lymphovascular invasion; PNI, perineural invasion; ENE, extranodal extension.

$p < 0.001$). The median PFS was longer in patients without prior CRT (6 months vs. 4 months, $p = 0.046$). Other factors including chemotherapy regimen (PF or taxane-based), chemotherapy dose (PF dose), or CRT refraction status did not lead to significant effect on PFS. In regard to analysis of OS, the patients who received more cetuximab treatment ($\geq 11$ cycles) had a better median OS than those who received less cetuximab (12.43 months vs. 4.46 months, $p < 0.001$). Other factors, including chemotherapy regimen and dose, did not lead to significant effects on PFS. The OS curves are shown in Fig. 3.

**Table 2   Treatment modality.**

| Variables | n (%) |
|---|---|
| Previous treatment | |
|    Surgery | 83 (78.3%) |
|    Chemotherapy | 86 (81.1%) |
|    CRT | 85 (80.2%) |
| CRT-refractory | 34 (32.1%) |
| Cetuximab applied reason | |
|    Metastasis | 65 (61.3%) |
|    Recurrence | 41 (38.7%) |
| Cetuximab cycle, median (range) | 11 (2-24) |
|    <11 | 46 (43.4%) |
|    $\geq 11$ | 60 (56.6%) |
| Regimen of chemotherapy | |
|    PF | 76 (71.7%) |
|    Taxane-based | 17 (16.0%) |
|    Others | 13 (12.3%) |
| Platinum | |
|    Cisplatin | 85 (80.2%) |
|    Carboplatin | 5 (4.7%) |
| Chemotherapy dose | |
|    60/800 | 36 (34.0%) |
|    75/1000 | 57 (53.8%) |
| Disease progressed | 105 (99.1%) |
| ORR | 30 (28.3%) |
| DCR | 51 (48.1%) |
| Median PFS (months, 95% CI) | 5.00 (3.00–6.00) |
| All-cause mortality | 68 (64.2%) |
| Median OS (months, 95% CI) | 9.23 (7.03–13.84) |

**Notes.**
CRT, concurrent chemoradiotherapy; PF, cisplatin and fluorouracil; ORR, overall response rate; DCR, disease control rate; PFS, progression-free survival; OS, overall survival; 95% CI, 95% confidence intervals.

Next, we applied a landmark method for further validation. Because responses could be observed within the first 3 months following cetuximab exposure, a 3-month landmark was used. After excluding patients who progressed or died within the three months, the patients with more cycles of cetuximab ($\geq 11$ cycles) still showed better median PFS (8 months vs. 2 months, $p = 0.057$) and OS (13.9 months vs. 5.07 months, $p = 0.0002$) than the patients treated with fewer cycles of cetuximab.

To clarify the effects of CRT-refraction on survival, we evaluated median PFS and OS in patients with or without CRT-refraction. In the non-CRT-refractory cohort ($n = 72$), the median PFS and OS were 5.00 months (95% CI [3.00–7.00]) and 10.43 months (95% CI [7.03–14.64]), respectively. The 3-year OS was 28.72% (95% CI [17.25–41.24]). On further evaluation of these 72 subjects, 27 patients with <11 cetuximab cycles obtained a 3-year PFS rate of 3.70% (95% CI [0.27–15.90]), and a 3-year OS rate of 2.22% (95% CI [0.18–10.15]).

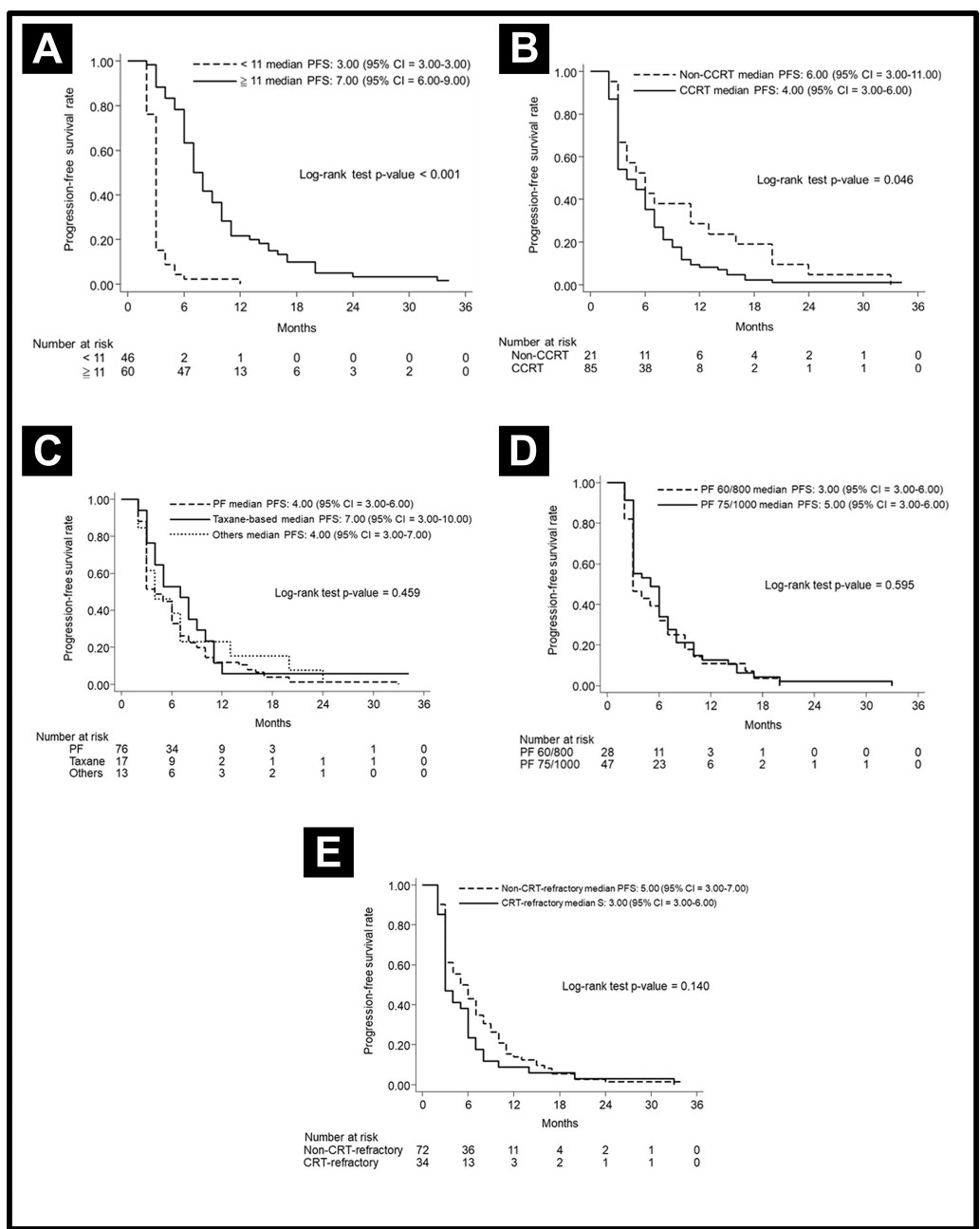

**Figure 2 Progression-free survival curve.** Progression-free survival curve according to (A) cetuximab cycle, (B) previous CRT, (C) different chemotherapy regimens, (D) different doses of PF, and (E) CRT-refractory patents or not.

Additionally, 45 patients with ≥ 11 cetuximab cycles obtained a 3-year PFS rate of 11.57% (95% CI [1.04–36.08]), and a 3-year OS rate of 37.07% (95% CI [21.60–52.59]). The patients treated with more cetuximab cycles also showed a better median PFS and OS then did the patients treated with fewer cetuximab cycles, shown in Fig. 4.

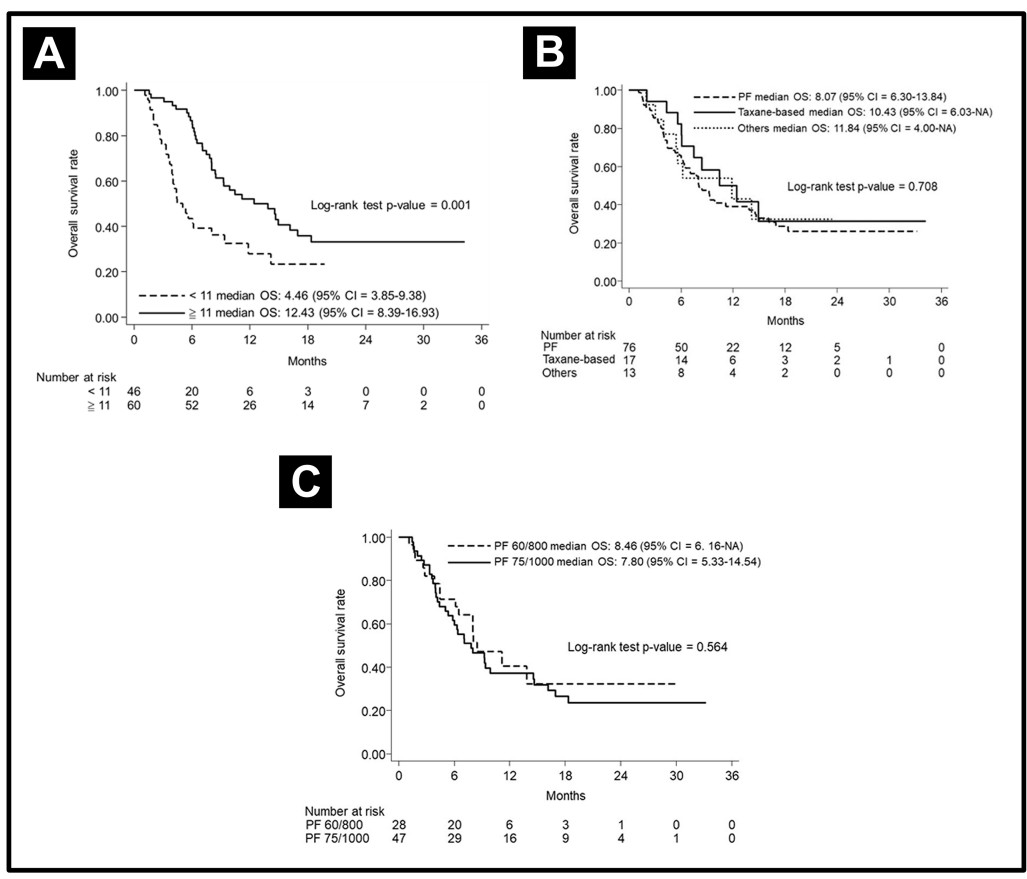

**Figure 3  Overall survival curve.** Overall survival curve according to (A) cetuximab cycle (B) different chemotherapy regimens, and (C) different doses of PF.

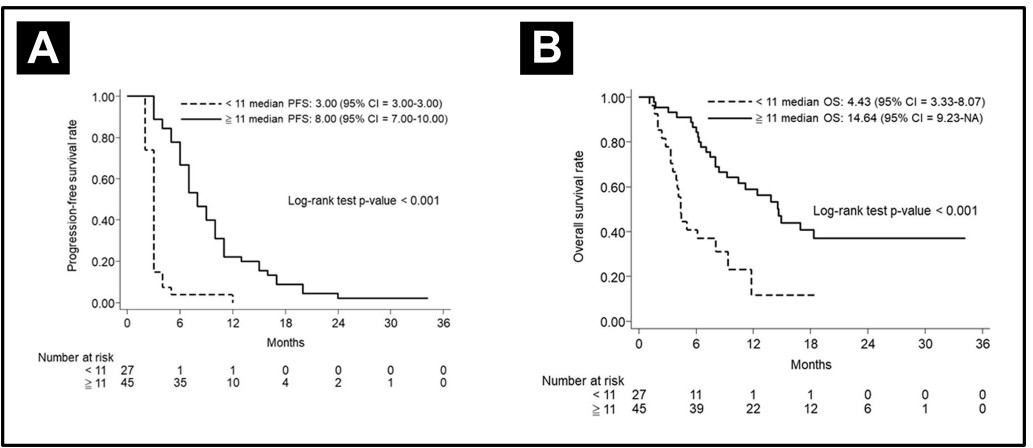

**Figure 4  Subgroups analysis in CRT-refractory patients.** (A) Progression-free survival curve and (B) Overall curve according to the cetuximab cycle in CRT-refractory patients.

In the CRT-refractory patients, the median PFS and OS were 3.00 months (95% CI [3.00–6.00]) and 7.8 months, respectively. The 3-year OS rate was 25.30% (95% CI [10.32–43.53]). Six CRT-refractory patients who used taxane-based regimens obtained a median PFS and OS of 3.00 months (95% CI [2.00–8.00]) and 5.62 months (95% CI [2.03–NA]), respectively. The 3-year OS was 16.67% (95% CI [0.77–51.68]).

### Risk factor investigation for disease progression

Risks of disease progression were analyzed using univariate regression consisting of parameters as age, alcohol, betel nuts, tobacco consumption, tumor site, margin positivity, histologic features (including lymphovascular invasion, perineural invasion, and extranodal extension), tumor size, lymph node status, stage, previous treatment modality (including surgery, chemotherapy, and CRT), treatment status, cetuximab cycles, dose, and regimens of chemotherapy. In addition, a subsequent multivariable regression analysis was performed to evaluate the significant progression factors in univariate analysis.

As shown in Table 3, positive perineural invasion was the independent factor related with shorter median PFS. N3 disease showed a trend toward poorer PFS ($p = 0.055$, univariate analysis). After adjustment for other different variables in the multivariable analysis, this difference became significant (HR $= 2.57$; $p = 0.043$). Significantly, treatment with more cetuximab cycles ($\geq 11$ cycles) was a favorable factor associated with better median PFS (HR $= 0.19$; $p < 0.001$, and HR $= 0.18$; $p < 0.001$ in univariate and multivariable analysis, respectively).

### Determining the risk factor for poorer overall survival

Similar clinicopathological factors were analyzed for overall survival. N2 disease had a significantly negative impact on OS (HR $= 2.09$; $P = 0.022$ and HR $= 4.79$; $p = 0.006$ in univariate and multivariable analyses, respectively). Treatment with more cetuximab cycles showed a significant, positive effect on OS (HR $= 0.46$; $p = 0.002$ and HR $= 0.48$; $p = 0.010$ in both univariate and multivariable analyses, respectively). Other factors with trends toward shorter OS included N3 disease ($p = 0.170$). After adjustment for other variables, this difference became significant in the multivariable analysis (HR $= 7.34$; $p = 0.005$). These results are shown in Table 4.

Although endemic habits showed no significant impact on PFS and OS, multiple endemic habits might increase risk in PFS and OS compared to single or double endemic habits. The impact of multiple endemic habits on PFS and OS are summarized in Table S4.

### Safety and tolerability

All grades and the worst grade 3 and grade 4 treatment-related adverse events (AEs) in patients receiving cetuximab therapy are listed in Table 5. Among the patients treated with the platinum/5FU and cetuximab regimen, the most common AEs were skin rash (2.6%), anemia (2.6%), neutropenia (1.3%), vomiting (1.3%) and fever (1.3%). Among patients treated with taxane-based regimens, only one patient suffered from grade 3 fever (5.9%). There were no grade 3 or grade 4 AEs in other groups. In general, skin rash was the most frequent cetuximab-related AE; however, most of patients tolerated it. There was no interstitial lung disease observed in our patients.

**Table 3  Cox regression for disease progression.**

| Variables | Comparison | Univariate | | Multivariable[a] | |
|---|---|---|---|---|---|
| | | HR (95% CI) | *P* | HR (95% CI) | *P* |
| Age | Years | 0.99 (0.97–1.01) | 0.502 | – | |
| Alcohol | Yes vs. no | 1.47 (0.88–2.44) | 0.141 | 1.47 (0.81–2.64) | 0.202 |
| Betel nuts | Yes vs. no | 1.17 (0.67–2.05) | 0.578 | – | |
| Smoking | Yes vs. no | 0.92 (0.50–1.69) | 0.783 | – | |
| Histology | OC vs. HPC | 1.32 (0.81–2.17) | 0.270 | – | |
| | OPC vs. HPC | 0.95 (0.49–1.83) | 0.871 | – | |
| Margin | With vs. without residual tumor | 1.30 (0.67–2.51) | 0.442 | – | |
| Grade | 2 vs. 1 | 0.87 (0.55–1.38) | 0.563 | – | |
| | 3 vs. 1 | 1.03 (0.56–1.91) | 0.920 | – | |
| LVI | Positive vs. negative | 2.04 (0.69–6.02) | 0.195 | 0.43 (0.11–1.72) | 0.231 |
| PNI | Positive vs. negative | **2.89 (1.26–6.65)** | **0.012** | **3.19 (1.08–9.46)** | **0.036** |
| ENE | Positive vs. negative | 1.18 (0.38–3.61) | 0.776 | – | |
| Tumor size | T1 vs. T0 | 0.19 (0.04–0.85) | 0.029 | 0.75 (0.14–3.96) | 0.739 |
| | T2 vs. T0 | 0.29 (0.07–1.28) | 0.102 | 0.78 (0.16–3.75) | 0.751 |
| | T3 vs. T0 | 0.41 (0.09–1.83) | 0.244 | – | |
| | T4 vs. T0 | 0.27 (0.06–1.13) | 0.073 | 0.82 (0.17–3.89) | 0.805 |
| Lymph node status | N1 vs. N0 | 1.19 (0.60–2.37) | 0.620 | – | |
| | N2 vs. N0 | 1.73 (1.06–2.81) | 0.027 | 1.85 (0.98–3.51) | 0.059 |
| | N3 vs. N0 | 2.04 (0.98–4.24) | 0.055 | **2.57 (1.03-6.43)** | **0.043** |
| Stage | II vs. I | 1.66 (0.59–4.69) | 0.339 | – | |
| | III vs. I | 1.76 (0.72–4.28) | 0.214 | – | |
| | IV vs. I | 1.50 (0.75–3.02) | 0.252 | – | |
| Surgery | With vs. without | 0.80 (0.50–1.28) | 0.354 | – | |
| Chemotherapy before target therapy | With vs. without | 0.87 (0.53–1.42) | 0.585 | – | |
| CRT-refractory | Yes vs. no | 1.32 (0.87–1.99) | 0.191 | 1.18 (0.72–1.91) | 0.511 |
| Cetuximab applied reason | Metastasis vs. recurrence | 1.002 (0.68–1.49) | 0.992 | – | |
| Cetuximab cycle, median (range) | ≥11 vs. <11 | **0.19 (0.11–0.30)** | **<0.001** | **0.18 (0.09–0.33)** | **<0.001** |
| Regimen of chemotherapy | Taxane-based vs. PF | 0.75 (0.44–1.29) | 0.297 | – | |
| | Others vs. PF | 0.85 (0.47–1.54) | 0.591 | – | |
| Platinum | Carboplatin vs. Cisplatin | 0.55 (0.22–1.39) | 0.206 | – | |
| Chemotherapy dose | 75/1000 vs. 60/800 | 0.90 (0.56–1.43) | 0.644 | – | |

**Notes.**

HPC, hypopharyngeal cancer; OC, oral cavity cancer; OPC, oropharyngeal cancer; LVI, lymphovascular invasion; PNI, perineural invasion; ENE, extranodal extension; CRT, concurrent chemoradiotherapy; PF, cisplatin and fluorouracil; HR, hazard ratio; 95% CI, 95% confidence intervals.

[a]Variables with *p*-value less than 0.2 in univariate analysis were included in multivariable model.

# DISCUSSION

The treatment options for HNSCC are sophisticated and require multidisciplinary groups to tailor personalized treatment. Since 2008, the addition of cetuximab to chemotherapy has become the first-line treatment of RM HNSCC regarding advancements in response and survival (*Vermorken et al., 2008*). However, HNSCC is a heterogenous disease and considerable effects of carcinogens have been reported, especially in the Asian population

**Table 4  Cox regression for overall mortality.**

| Variables | Comparison | Univariate | | Multivariable[a] | |
|---|---|---|---|---|---|
| | | HR (95% CI) | P | HR (95% CI) | P |
| Age | Years | 1.004 (0.98–1.03) | 0.738 | – | |
| Alcohol | Yes vs. no | 1.87 (0.95–3.67) | 0.070 | 2.00 (0.94–4.26) | 0.073 |
| Betel nuts | Yes vs. no | 1.50 (0.74–3.04) | 0.260 | – | |
| Smoking | Yes vs. no | 0.72 (0.37–1.42) | 0.341 | – | |
| Histology | OC vs. HPC | 1.41 (0.76–2.64) | 0.278 | – | |
| | OPC vs. HPC | 1.44 (0.67–3.12) | 0.350 | – | |
| Margin | With vs. without residual tumor | 0.86 (0.40–1.86) | 0.703 | – | |
| Grade | 2 vs. 1 | 0.91 (0.52–1.60) | 0.737 | – | |
| | 3 vs. 1 | 1.16 (0.57–2.36) | 0.672 | – | |
| LVI | Positive vs. negative | 1.89 (0.62–5.78) | 0.266 | – | |
| PNI | Positive vs. negative | 1.92 (0.76–4.88) | 0.169 | 0.54 (0.16–1.80) | 0.318 |
| ENE | Positive vs. negative | 0.92 (0.27–3.14) | 0.890 | – | |
| Tumor size | T1 vs. T0 | 0.05 (0.01–0.27) | <0.001 | 0.10 (0.01–1.13) | 0.063 |
| | T2 vs. T0 | 0.07 (0.02–0.36) | 0.001 | 0.14 (0.02–1.02) | 0.052 |
| | T3 vs. T0 | 0.06 (0.01–0.33) | 0.001 | 0.21 (0.02–1.73) | 0.145 |
| | T4 vs. T0 | 0.08 (0.02–0.35) | 0.001 | 0.26 (0.03–2.01) | 0.198 |
| Lymph node status | N1 vs. N0 | 1.59 (0.63–4.00) | 0.322 | 3.09 (0.72–13.16) | 0.128 |
| | N2 vs. N0 | **2.09 (1.11–3.92)** | **0.022** | **4.79 (1.55–14.77)** | **0.006** |
| | N3 vs. N0 | 1.92 (0.76–4.88) | 0.170 | **7.34 (1.85–29.16)** | **0.005** |
| Stage | II vs. I | 2.75 (0.79–9.51) | 0.110 | 1.69 (0.19–15.31) | 0.640 |
| | III vs. I | 0.85 (0.23–3.18) | 0.812 | 0.15 (0.02–1.42) | 0.098 |
| | IV vs. I | 1.56 (0.62–3.91) | 0.341 | 0.14 (0.02–1.08) | 0.060 |
| Surgery | With vs. without | 0.66 (0.38–1.13) | 0.127 | 0.83 (0.46–1.51) | 0.541 |
| Chemotherapy before target therapy | With vs. without | 1.25 (0.64–2.46) | 0.517 | – | |
| CRT-refractory | Yes vs. no | 1.20 (0.73–1.98) | 0.479 | – | |
| Cetuximab applied reason | Metastasis vs. recurrence | 1.16 (0.70–1.91) | 0.561 | – | |
| Cetuximab cycle, median (range) | ≥11 vs. <11 | **0.46 (0.28–0.75)** | **0.002** | **0.48 (0.27–0.84)** | **0.010** |
| Regimen of chemotherapy | Taxane-based vs. PF | 0.75 (0.38–1.49) | 0.417 | – | |
| | Others vs. PF | 0.90 (0.43–1.89) | 0.777 | – | |
| Platinum | Carboplatin vs. Cisplatin | 0.51 (0.16–1.64) | 0.260 | – | |
| Chemotherapy dose | 75/1000 vs. 60/800 | 1.19 (0. 66–2.17) | 0.564 | – | |

**Notes.**

HPC, hypopharyngeal cancer; OC, oral cavity cancer; OPC, oropharyngeal cancer; LVI, lymphovascular invasion; PNI, perineural invasion; ENE, extranodal extension; CRT, concurrent chemoradiotherapy; PF, cisplatin and fluorouracil; HR, hazard ratio; 95% CI, 95% confidence intervals.

[a]Variables with p-value less than 0.2 in univariate analysis were included in multivariable model.

(*Network, 2015*). Accessibility to expensive drugs and restrictions on reimbursement policies also have impacts on the responses and outcomes of treatment in many countries, including Taiwan (*Davidoff et al., 2018*; *Hsu, Wei & Yang, 2019*; *Morgan & Kennedy, 2010*). This retrospective study highlights the important role of cetuximab cycles in RM HNSCC, especially in an endemic carcinogen exposure area such as Taiwan.

In this study, 106 patients treated with cetuximab-based regimens were assessed; most patients had the habit of using an addictive substance and over half the patients had

**Table 5  Adverse effects observed according to CTCAE version 4.0.**

| | PF | | | | Taxane-based | | | | Others | | | |
|---|---|---|---|---|---|---|---|---|---|---|---|---|
| | All grades | | Grade 3–4 | | All grades | | Grade 3–4 | | All grades | | Grade 3–4 | |
| | No. | % | No. | % | No. | % | No. | % | No. | % | No. | % |
| Febrile | 7 | 9.2 | 1 | 1.3 | 4 | 23.5 | 1 | 5.9 | 2 | 15.4 | 0 | – |
| Neutropenia | 24 | 31.6 | 1 | 1.3 | 6 | 35.3 | 0 | – | 2 | 15.4 | 0 | – |
| Skin rash | 46 | 60.5 | 2 | 2.6 | 9 | 52.9 | 0 | – | 5 | 38.5 | 0 | – |
| Anemia | 51 | 67.1 | 2 | 2.6 | 14 | 82.4 | 0 | – | 4 | 30.8 | 0 | – |
| Hypomagnesemia | 31 | 40.8 | 0 | – | 11 | 64.7 | 0 | – | 4 | 30.8 | 0 | – |
| Pneumonia | 7 | 9.2 | 0 | – | 2 | 11.8 | 0 | – | 1 | 7.7 | 0 | – |
| Infusion reaction | 5 | 6.6 | 0 | – | 0 | – | 0 | – | 0 | – | 0 | – |
| Vomiting | 28 | 36.8 | 1 | 1.3 | 5 | 29.4 | 0 | – | 8 | 61.5 | 0 | – |

concurrent exposure to all three addictive substances. However, our outcomes were not inferior when indirectly compared to those of other clinical trials, including the EXTREME regimen conducted by European cancer institutes (*De Mello et al., 2014*) and the EXTREME trial (*Vermorken et al., 2008*). The possible reasons may relate to regular and frequent follow-up, laboratory, and imaging studies to detect disease progression and guide subsequent treatment plan when progression was noted. Compared to the aforementioned Asian trial, including Japanese (*Tahara et al., 2016*) and Chinese trials (*Guo et al., 2015*), the ORR of our study was slightly lower, which may be related to usage of cetuximab maintenance, different regimens of chemotherapy, and a patient population with distinct endemic carcinogen exposures. The patients in the Japanese trial received cetuximab maintenance and chemotherapy with carboplatin and paclitaxel. However, there was nearly no effect of betel nuts in the Japanese population. The effects of carcinogen were also not mentioned in the Chinese and Korean population. The results of these studies are summarized in Table 6 (*Adkins et al., 2018*; *Bossi et al., 2017*; *De Mello et al., 2014*; *Friesland et al., 2018*; *Guigay et al., 2016*; *Guigay et al., 2012*; *Guigay et al., 2019*; *Guo et al., 2015*; *Tahara et al., 2016*; *Vermorken et al., 2008*).

Importantly, the median PFS and OS of our study are compatible with those of another retrospective study (*De Mello et al., 2014*). Our real-world results were also comparable with those of other clinical trials. As we mentioned, these may be related to every diagnosed patient receiving frequent physical and imaging examinations, receiving care from a multidisciplinary team (including nurse case management, integrating expertise of medical oncologist, surgeon, radiologists, case managers, nurses, nutritionists, and pharmacists), and meeting periodically to discuss treatment direction, evaluating therapeutic effects, and providing further recommendations. As noted in breast cancer care, earlier detection from more aggressive monitoring could lead to improved treatment strategies and possibly improved survival (*Graham et al., 2014*).

Although our study was conducted retrospectively in a single medical center, our study reflects the observation of the real-world setting in an endemic carcinogen exposure area. However, our study still had limitations in terms of relatively smaller sample size

Wang et al. (2020), *PeerJ*, DOI 10.7717/peerj.9862

**Table 6 Comparisons between different trials of cetuximab-based chemotherapy.**

| Study | Country | Year | Author | Chemotherapy | Cetuximab maintenance | Numbers | ORR (%) | OS (m) |
|---|---|---|---|---|---|---|---|---|
| Extreme | Belgium | 2008 | Vermorken JB | Cisplatin 100 mg/m2 D1 Fluorouracil 1000 mg/m2 D1-4 Q3W | Weekly | 222 | 36 | 10.1 |
| GORTEC 2008-03 | France and Belgium | 2012 | Guigay J | Cisplatin 75 mg/m2 D1 Docetaxel 75 mg/m2 D1 Q3W | Biweekly | 54 | 44 | 14 |
| NCT01177956 | China and South Korea | 2014 | Guo Y | Cisplatin 75 mg/m2 Fluorouracil 750 mg/m2 D1-5 Q3W | Weekly | 68 | 55.9 | 12.6 |
| CET-INT | Italy | 2017 | Bossi P | Cisplatin 75 mg/m2 D1 Paclitaxel 175 mg/m2 D1 Q3W | Weekly | 191 | 51.7 | 11 |
| CSPRO-HN02 | Japan | 2016 | Tahara M | Carboplatin AUC 2.5 D1, D8 Paclitaxel 100 mg/m2 D1, D8 Q3W | Weekly | 47 | 40 | 14.7 |
| CACTUX | USA | 2018 | Adkins D | *nab*-paclitaxel 100 mg/m2 weekly Carboplatin AUC 5 D1 or Cisplatin 75 mg/m2 D1 Q3W | Weekly | 32 | 63 | 18.8 |
| CETMET | Demark | 2018 | Friesland S | Cisplatin 75 mg/m2 D1 Paclitaxel 175 mg/m2 D1 Q3W | Biweekly | 85 | 63 | 10.2 |
| TPEx | France and Belgium | 2019 | Guigay J | Cisplatin 75 mg/m2 D1 Docetaxel 75 mg/m2 D1 Q3W | Biweekly | 269 | 46 | 14.5 |
| Real world practice | European | 2014 | De Mello RA | Cisplatin 100 mg/m2 D1 Fluorouracil 1000 mg/m2 D1-4 Q3W | Weekly | 121 | 23.91 | 11 |
| Real world practice | Taiwan | 2020 | Wang | Cisplatin 75 mg/m2 D1 Fluorouracil 1000 mg/m2 D1-4 Q3W | No | 106 | 28.3 | 9.23 |

**Notes.**

ORR, overall response rate; OS, overall survival; Q3W, every three weeks; AUC, area under the curve.

and inevitable time bias. To address the immortal time bias and reverse causality, we applied landmark analysis, which suggested more cycles of cetuximab may bring survival benefit to HNSCC patients. The heterogeneous study population is also an issue. Unlike the EXTREME or TPEX studies that excluded CRT-refractory patients, we included CRT-refractory patients. Furthermore, patients who received nonplatinum chemotherapy regimens, including taxane and MTX, were also included. Heterogeneity of the study population may confound the analysis. However, our findings revealed real-world conditions in term of financial burden of novel treatment, which lead to absence of cetuximab maintenance. In addition, our study included a Taiwanese population with high incidence of oral cavity cancer that may be related to strong carcinogen exposure, including alcohol, betel nuts, and tobacco. Previous studies had revealed lower expression of tumor suppressor gene p53 alterations, higher percentage of MDM2 protein expression, as well as higher rate of Ras oncogene mutation after long-term exposure to betel nuts (*Huang et al., 2001*; *Kuo et al., 1994*; *Kuo et al., 1999*). The upregulation of *EGFR* has been confirmed in betel-nut-associated cancer of the oral cavity associated with poor prognosis (*Sheu et al., 2009*). Three amplicons (KRAS, MAPK1, and CCND1) have been observed in cancer of oral cavity from Taiwanese patients, and therefore, all could possibly contribute to activation of EGFR signaling (*Sheu et al., 2009*). EGFR protein upregulation, excluding the effect of *EGFR* gene copy number on protein overexpression, was related to poor differentiation of tumor cells and lymph node metastasis, especially extranodal extension (*Huang et al., 2017*). Taken together, cetuximab targeting EGFR on HNSCC cells induces potent antibody-dependent cell-mediated cytotoxicity that further augments anti-tumor effect when combined with chemotherapy (*Specenier & Vermorken, 2013*).

The restrictions in targeted therapy-related reimbursement policies defer patients' benefits related to RM HNSCC. The limitation of a total 18 cycles of cetuximab without maintenance has been in place since 2016 in Taiwan. In other countries, cetuximab maintenance plays an important role in improving survival and outcomes with tolerable adverse events (*Wakasugi et al., 2015*). The median duration of maintenance was 11 weeks in the EXTREME trial, 16 weeks in a real-world study in France, and 17 weeks in a real-world study in Portugal. Broadening the duration of the eligible patient population to targeted therapies may be an effective way to improve clinical outcomes of treatments.

## CONCLUSIONS

Consistent administration of cetuximab provides potential clinical benefits in HNSCC patients in endemic carcinogen exposure areas in an Asian population; therefore, longer cetuximab maintenance therapy is urgently warranted in these patients with poor prognoses.

### Funding

This work was supported by the following grants: KMUH107-7M12, KMUH108-8R23, KMUH108-8M12, and KMHK-DK109004 from the Kaohsiung Medical University Hospital. The funders had no role in study design, data collection and analysis, decision to publish, or preparation of the manuscript.

### Grant Disclosures

The following grant information was disclosed by the authors:
Kaohsiung Medical University Hospital: KMUH107-7M12, KMUH108-8R23, KMUH108-8M12, KMHK-DK109004.

### Competing Interests

The authors declare there are no competing interests.

### Author Contributions

- Hui-Ching Wang conceived and designed the experiments, performed the experiments, prepared figures and/or tables, authored or reviewed drafts of the paper, and approved the final draft.
- Pei-Lin Liu, Pei-Chuan Lo and Yi-Tzu Chang performed the experiments, analyzed the data, prepared figures and/or tables, authored or reviewed drafts of the paper, and approved the final draft.
- Leong-Perng Chan conceived and designed the experiments, performed the experiments, prepared figures and/or tables, and approved the final draft.
- Tsung-Jang Yeh conceived and designed the experiments, analyzed the data, prepared figures and/or tables, and approved the final draft.
- Hui-Hua Hsiao analyzed the data, authored or reviewed drafts of the paper, and approved the final draft.
- Shih-Feng Cho conceived and designed the experiments, analyzed the data, authored or reviewed drafts of the paper, and approved the final draft.

### Ethics

The following information was supplied relating to ethical approvals (i.e., approving body and any reference numbers):

This study was approved by the Institutional Review Board and Ethics Committee of Kaohsiung Medical University Hospital (KMUHIRB-E(II)-20190357).

### Data Availability

Data are available as a Supplemental File.

### Supplemental Information

Supplemental information for this article can be found online at http://dx.doi.org/10.7717/peerj.9862#supplemental-information.

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
