# Peer review of "Consistent administration of cetuximab is associated with favorable outcomes in recurrent/metastatic head and neck squamous cell carcinoma in an endemic carcinogen exposure area: a retrospective observational study"

_PeerJ, doi:10.7717/peerj.9862_

## Round 0.1 · original submission · Major Revisions

Please respond to the reviewer's comments and make the suggested changes in your revised manuscript.

Reviewer 1 ·

Basic reporting

No comment

Experimental design

No comment

Validity of the findings

No comment

Additional comments

Overall, this is a well written article that clearly explains the important real-world data of cetuximab-containing treatment in RM HNSCC. Authors should consider providing high resolution images for all figures.

Reviewer 2 ·

Basic reporting

In their manuscript the authors retrospectively analyze the efficacy of cetuximab based therapy in 106 Taiwanese patients suffering from R/M HNSCC in a real-world setting. Since the outcome parameters were comparable to what has been published in the literature, the authors conclude that cetuximab is effective in this population as well.
Although it is a strength of this manuscript provide real-world data in a population with a high percentage of carcinogen exposure, a couple of questions arose, when reviewing the manuscript:
- The introduction section states that “oropharyngeal SCC accounts for the largest group of head and neck cancers related to HPV” (line 72) and that the most predominant site in Taiwan is OPC. However, it has to be noted that the most predominant HNSCC in western countries is OPC as well, which is usually HPV negative (see Cramer et al. Nature Rev.Cancer or for epidemiology data extraction from https://gco.iarc.fr/; or Gatta et al. EJC 2015 for Europe). Only a minority of the patients in the EXTREME trial were HPV positive (Vermorken, Annals of Oncology 2014). Thus, the introduction should be revised accordingly.
- The inclusion/exclusion criteria are poorly defined in the method section: Were patients included, which were platinum resistant? This seems to be likely, since patients resistant to CRT were included. Why were those patients not excluded? The EXTREME or TPEX trials excluded those patients. Why were patients included, who did not receive platinum based first line therapy such as MTX (line 145). This decision results in a heterogeneous population and a potential bias. The results are hardly comparable to the Extreme trial or the trials listed in table 5. All of them including the European real-world trial evaluated a well-defined homogenous population.
- Likewise the ORR is considerably lower in this study compared to the aforementioned trials and especially to the Japanese trial. This should be stated and discussed accordingly.
- It stated in the manuscript that cetuximab exposure <11 cycles is an independent risk factor for progression. However the reason for cetuximab discontinuation is not given. It is very likely that cetuximab was stopped because of disease progression. Therefore, disease progression might be the reason for cetuximab exposure <11 cycles or at least the very same endpoint and not the other way round?? What would figure 2A look like, when plotting the PFS curve in patients with at least SD vs. patients with PD?
- The adverse event rate should be included in a real-world report and compared to the literature. There are studies in HNSCC that show that cetuximab can cause severe lung injuries in Asian patients (Nakano et al, Head and neck 2019). Was ILD observed in this population?
- Additional ref. should be included and discussed such as Guo et al. Head and Neck 2014, who performed a cetuximab/platinum R/M HNSCC first line trial in a Chinese population.
- Revision with respect to English grammar is advised (i.e. line 322-324 is hardly comprehensible)

Experimental design

See above

Validity of the findings

See above

Additional comments

See above

·

Basic reporting

The first sentence of the ‘results’ section in the abstract does not add much value. It would rather be more interesting for the reader to get a sense of what proportion of total HNSCC patients developed recurrent or metastatic disease within the timeframe of the study.

Lines 85 – 87: It is misleading to suggest that Cetuximab is first-line therapy for RM HNSS across the board. The authors should clarify the rationale for choosing Cetuximab therapy when CPS <0

Line 99-102: Please fix grammar

Line 171 – multivariable

Line 197-198 – Please add percentage

Line 214: Table 2 does not summarize ORR. Have the authors missed including a table

Line 222: The ‘median’ PFS was calculated as 6 months. This has been overlooked in multiple places

Line 300: Not sure how the conclusion of more frequent visits can be drawn from this dataset

Experimental design

1) Line 124: Were interviews conducted in all patients in the cohort? If not, this could lead to information bias, and preferentially higher recall in these patients compared to those who were recruited on the basis of EMR alone.
2) Besides OS, and PFS, methods to assess categorical responses (CR, PR ,SD,PD) have not been detailed in the methods. Why was the RECIST methodology not used (PMC2785927, PMC5373019)? Additionally, was response determined at the time of first restaging or was this best objective response?
3) Line 160: Unclear to the reviewer which patients represent the control group? Please refrain from using ambiguous terminology - this is a retrospective cohort study
4) What was the median time interval between time of start of treatment and determination of response in all patients? What was the median number of cycles in all patients after which response was determined?
5) Line 169-170. Unclear what the rationale is for excluding patients who were alive and without disease progression at the time of last follow-up – this is not a typical censoring strategy. In fact, it would lead to overestimation of the true hazard ratio
6) Model building strategy is confusing. How were variables selected for inclusion in the final models? Additionally, multivariable models for both PFS and OS seem to be overfit, as there are likely more variables in the model than the number of events would allow. This reviewer recommends a best subsets or stepwise regression for more prudent model building

Validity of the findings

7) The results section is poorly written and is merely a line-by-line recapitulation of thee tables themselves. As such, the leader struggles with a lack of perspective. Along the same lines Tables 1 and 2 are largely uninformative. Since the authors seem to pivot around number of cycles of cetuximab CRT refraction, it would be interesting to see for instance what the distribution is of demographic characteristics is between >11 and <11 treatment cycles.
8) The use of a binary variable for CRT use, as well as a separate variable for CRT refraction seems redundant. This reviewer recommends using CRT refraction only. Additionally, a large majority of patients were Stage IV at initial diagnosis. Were the response rates better in patients with stage I – III at initial diagnosis? Why were earlier stage patients included in the analysis?

Additional comments

In their work entitled, ‘ Consistent administration of cetuximab is associated with favorable outcomes in recurrent/metastatic head and neck squamous cell carcinoma at endemic carcinogen exposure area: a retrospective observational study’, Wang et al. have studied factors associated with survival in a cohort of RM HNSCC patients treated with Cetuximab in a region of high endemic carcinogen exposure. The authors have specifically reported an association between CRT-refraction and number of Cetuximab cycles on treatment response. While the analysis itself is not novel, the choice of study population with endemic carcinogenic habits, and lack of drug accessibility is potentially interesting. That being said none of the endemic factors seem to impact survival in the cohort – in fact smoking seems to have a protective effect which is odd. Furthermore, several issues exist as have been detailed

---

## Round 0.2 · Minor Revisions

Although you have made major changes that were mostly acceptable to the reviewers, there are still some issues that need to addressed in your manuscript. Please address the issues raised by the reviewers and submit your revised manuscript.

Reviewer 2 ·

Basic reporting

Although the authors addressed the majority of the issues stated, there are still a few issues pending:
-Abstract: A “conclusion section” is missing.
- it is still incorrect that “In western countries, oropharyngeal SCC accounts for the largest group of HNSCC”, since oral cavity is the largest group.
- Revision with respect to English grammar is still advised for the whole manuscript.

Experimental design

See above

Validity of the findings

See above

Additional comments

-

·

Basic reporting

Please see comments to author

Experimental design

Please see comments to author

Validity of the findings

Please see comments to author

Additional comments

This reviewer appreciates the authors rebuttal and most issues that were raised have been adequately addressed. The flow of the manuscript is also much better. A few outstanding issues still remain that requires their attention, though missing line numbers made it difficult to recommend specific changes.

• Several grammatical errors still exist, and a thorough proof-read is imminently warranted
• Please replace multivariate with multivariable throughout the manuscript. The difference is key. Multivariate refers to multiple outcome which is not the case for this analysis
• Per the letter – line 99-102 is still grossly incorrect – “Different from clinical trials…”. Please fix
• Median time to determination of response should be included in the manuscript
• Regarding the impact of endemic habits, what is the impact of OS in patients who have more than 1 endemic habit compared to those who have multiple endemic habits? For ex: smoking alone vs smoke + alcohol use?
• Along the same line, the authors belief in the novelty of this work lies in the population being ‘endemic’. Perhaps a paragraph is warranted in the discussion section to go over how they are impacting treatment responses.
• Study design: It would be important to clarify that the primary endpoint was assessed after treatment with cetuximab
• Table 2:
Were SD patients also included in the calculation of ORR? If yes, how long were they classified as SD
What was the median time to disease progression?
Since 99.1% of patients progressed, how was DCR defined? No description of this estimation has been provided in the methods
• Interesting that there are no differences in concurrent chemotherapy use between patients who received <11 vs >11 cycles. This would be an important fact to highlight
• Please refrain from using the word ‘favorite’ in describing the predictor variable after multivariable adjustment
• Table 3: Why suddenly change the annotation from ‘Cetuximab’ to ‘Erbitux’. Please be consistent

---

## Round 0.3 · accepted · Accept

Thanks for addressing the issues raised by the reviewers in your revised manuscript.